# Registered report: IDH mutation impairs histone demethylation and results in a block to cell differentiation

Adam D Richarson[1], David A Scott[1], Olga Zagnitko[1], Pedro Aza-Blanc[2], Chih-Cheng Chang[2], David A Russler-Germain[3], Reproducibility Project: Cancer Biology*

[1]Cancer Metabolism Facility, Sanford Burnham Prebys Medical Discovery Institute, La Jolla, United States; [2]Functional Genomics Core, Sanford Burnham Prebys Medical Discovery Institute, La Jolla, United States; [3]Washington University School of Medicine, St Louis, United States

**\*For correspondence:** fraser@ scienceexchange.com

**Group author details:** Reproducibility Project: Cancer Biology See page 23

**Abstract** The Reproducibility Project: Cancer Biology seeks to address growing concerns about reproducibility in scientific research by conducting replications of selected experiments from a number of high-profile papers in the field of cancer biology. The papers, which were published between 2010 and 2012, were selected on the basis of citations and Altmetric scores (Errington et al., 2014). This Registered Report describes the proposed replication plan of key experiments from "IDH mutation impairs histone demethylation and results in a block to cell differentiation" by Lu and colleagues, published in Nature in 2012 (Lu et al., 2012). The experiments that will be replicated are those reported in Figures 1B, 2A, 2B, 2D and 4D. Lu and colleagues demonstrated that expression of mutant forms of IDH1 or IDH2 caused global increases in histone methylation and increased levels of 2 hydroxyglutarate (Figure 1B). This was correlated with a block in differentiation (Figures 2A, B and D). This effect appeared to be mediated by the histone demethylase KDM4C (Figure 4D). The Reproducibility Project: Cancer Biology is a collaboration between the Center for Open Scienceand Science Exchange, and the results of the replications will be published by *eLife*.

## Introduction

Mutations in the metabolic proteins IDH1 and IDH2 are associated with gliomas, acute myeloid leukemias, chondrosarcomas, intrahepatic cholangiocarcinomas, lymphomas, melanomas and colon, thyroid and prostate cancers (for review, see *Krell et al., 2013*). Previous work has shown that these mutations change the specificity of the reaction catalyzed by IDH proteins; instead of producing α-ketoglutarate from isocitrate, they produce 2-hydroxyglutarate (2HG), a metabolite that can have oncogenic effects (*Krell et al., 2013*; *McKenney and Levine, 2013*; *Ward et al., 2010*; *Xu et al., 2011*; *Zhang et al., 2013*). Lu and colleagues expand upon this work to identify a potential mechanism for how 2HG can effect major changes in cell behavior. They present evidence that 2HG interferes with global demethylation that is required for progenitor cells to complete terminal differentiation. Transfection of 3T3-L1 cells with the mutant forms of *IDH1* and *IDH2* that produce 2HG lead to an increase in global methylation levels and prevented normal in vitro differentiation into adipocytes. The 2HG-sensitive histone demethylase KDM4C appeared to be required for this process, as knockdown of KDM4C recapitulated the phenotype of 2HG production. Examination of glioma samples showed a correlation between *IDH* mutation status and level of overall methylation

(*Lu et al., 2012*). Taken together, Lu and colleagues' findings help explain how mutations in IDH1 and IDH2 potentially interface with cancer development and progression.

In Figure 1B, Lu and colleagues examined the effects of mutations in *IDH1* and *IDH2* on global levels of methylation by transfecting mutant and wild type forms of the genes into 293T cells and using Western blot to assess the levels of various methylation markers. They also confirmed that introduction of the mutated forms of *IDH1* and *IDH2* correlated with increased intracellular levels of the oncometabolite 2HG. Their findings suggest that mutations in *IDH1* and *IDH2* correlate with increased levels of many methylation markers, and this key finding is replicated in Protocol 1.

In order to understand the effects of hypermethylation more fully, Lu and colleagues turned to an in vitro model of differentiation; when treated with appropriate signals, 3T3-L1 cells undergo epigenetic changes required for them to differentiation into adipocytes. In Figure 2A and B, they transfect undifferentiated 3T3-L1 cells with the wild type and mutant forms of IDH2 and assess the cells' ability to differentiate into adipocytes, as determined by staining for lipid droplets with Oil-Red-O. While differentiated 3T3-L1 cells transfected with vector only or wild type *IDH2* showed robust Oil-Red-O staining, cells transfected with mutant *IDH2* did not, indicating a block in differentiation. qRT-PCR confirmed that cells transfected with mutant *IDH* variants did not express high levels of known adipocyte markers (Figure 2D). This key finding will be replicated in Protocol 2.

Lu and colleagues identified a histone demethylase, *KDM4C*, expressed as 3T3-L1 differentiation progressed, that appeared to be sensitive to 2HG. In Figure 4D, they use siRNAs to knock down levels of *KDM4C* in differentiating 3T3-L1 cells. Western blot analysis and Oil-Red-O staining confirmed that loss of KDM4C increased global methylation levels and inhibited differentiation. This key finding will be replicated in Protocol 3.

Several aspects of Lu's findings have been corroborated by other work. Multiple groups have demonstrated that perturbations in IDH proteins alter methylation levels; overexpression of the $IDH1^{R132H}$ allele in human tumor cells lines increased global histone methylation levels (*Duncan et al., 2012*), exogenous $IDH2^{R140Q}$ increased methylation levels in erythroleukemia progenitor cells (*Kernytsky et al., 2015*) and an immortalized astrocyte cell line expressing $IDH1^{R132H}$ also demonstrated increased levels of methylation (*Turcan et al., 2012*). Members of the Thompson lab (authors of this study) have confirmed that expression of mutant variants of IDH proteins in 3T3-L1 cells blocked differentiation into adipocytes (*Londono Gentile et al., 2013*; *Ward et al., 2013*). Sasaki and colleagues have shown that mutant *IDH1* expression increased levels of methylation in mice (*Sasaki et al., 2012*), while Akbay and colleagues published a similar observation for mutant forms of *IDH2* (*Akbay et al., 2014*). This effect may even hold true for human patients, as there is a marked increase in H3K9me3 levels associated with *IDH* mutations in oligodendromas and high grade astrocytomas (*Venneti et al., 2013*).

## Materials and methods

Unless otherwise noted, all protocol information was derived from the original paper, references from the original paper, or information obtained directly from the authors. An asterisk (*) indicates data or information provided by the Reproducibility Project: Cancer Biology core team. A hashtag (#) indicates information provided by the replicating lab.

### Protocol 1: Assessing the methylation status and 2HG production of 293T cells transfected with mutant forms of IDH1 and IDH2

This protocol describes how to transfect 293T cells with wild-type and mutant forms of IDH1 and IDH2 and assess levels of global methylation and 2HG production, as seen in Figure 1B and Supplemental Figure 1.

#### Sampling

- Experiment will be repeated a total of 6 times for a minimum power of 80%. The metabolite data is qualitative, thus to determine an appropriate number of replicates to initially perform, sample sizes based on a range of potential variance was determined.
  - See "power alculations' for details.
  - The metabolite data displayed in the bottom of Figure 1B were derived from Figure 3B of Figueroa and colleagues (*Figueroa et al., 2010*).

- Each experiment consists of five cohorts:
  - Cohort 1: 293T cells transfected with vector only
  - Cohort 2: 293T cells transfected with wild-type *IDH1*
  - Cohort 3: 293T cells transfected with *IDH1*^R132H
  - Cohort 4: 293T cells transfected with wild type *IDH2*
  - Cohort 5: 293T cells transfected with *IDH2*^R172K
- Each cohort is then examined for methylation status by Western blot and levels of 2HG production by GC-MS.

## Materials and reagents

| Reagent | Type | Manufacturer | Catalog # | Comments |
|---|---|---|---|---|
| 10 cm tissue culture dishes | Labware | Thermo Scientific | 130182 | Original unspecified |
| Bradford Assay Kit | Reporter assay | Bio-Rad | 500-0201EDU | Original unspecified |
| DMEM | Cell culture | Corning | 15013 CV | Replaces original from Invitrogen |
| Endo-free plasmid maxiprep kit | Kit | Qiagen | 12362 | Original unspecified |
| Fetal bovine serum (FBS) | Cell culture | CellGro | 10437-028 | Original cat # unspecified |
| 293T cells | Cell line | ATCC | CRL-3216 | Original source unspecified |
| HRP-conjugated donkey anti-rabbit secondary | Antibody | GE Healthcare | NA934V | |
| HRP-conjugated sheep anti-mouse secondary | Antibody | GE Healthcare | NA931V | |
| *IDH1* ORF clone | Nucleic acid | OriGene | RC210582 | Replaces ATCC plasmid in pCMV-Sport6 |
| *IDH1*^R132H ORF clone | Nucleic acid | OriGene | RC400096 | Original generated by authors |
| *IDH2* ORF clone | Nucleic acid | OriGene | RC201152 | Replaces Invitrogen plasmid in pOTB7 |
| *IDH2*^R172K ORF clone | Nucleic acid | OriGene | RC400103 | Original generated by authors |
| Mouse IgG1 monoclonal anti-IDH2 | Antibody | Abcam | Ab55271 | |
| Nitrocellulose membrane | Western blot reagent | Life Technologies | LC2006 | Original source unspecified |
| Nonfat milk | Western blot reagent | Carnation | | Original source unspecified |
| NuPAGE 4-12% precast gradient gel | Western blot reagent | Invitrogen | WG1401BOX | Original source unspecified |
| Pierce™ ECL Plus Western Blotting Substrate | Western blot reagent | Life Technologies | 32132 | Original unspecified |
| pLPC vector plasmid (pLPC H-Ras V12) | Nucleic acid | Addgene | 18741 | Original source unspecified |
| Ponceau stain | Chemical | SIGMA | P7170-1L | Original unspecified |
| Protease Inhibitor Cocktail, | Inhibitor | Sigma-Aldrich | P8340 | Original unspecified |
| Rabbit IgG monoclonal anti-H3 | Antibody | Cell Signaling Technology | 4499 | |
| Rabbit monoclonal anti-H3K4me3 | Antibody | Millipore | 17-614 | |
| Rabbit polyclonal anti-H3K36me3 | Antibody | Abcam | Ab9050 | |
| Rabbit polyclonal anti-H3K79me2 | Antibody | Cell Signaling Technology | 9757 | |

*Continued on next page*

*Continued*

| Reagent | Type | Manufacturer | Catalog # | Comments |
|---|---|---|---|---|
| Rabbit polyclonal anti-H3K9me2 | Antibody | Cell Signaling Technology | 9753 | |
| Rabbit polyclonal anti-H3K9me3 | Antibody | Abcam | Ab8898 | |
| Rabbit polyclonal anti-IDH1 | Antibody | ProteinTech | 12332-1-AP | |
| TBS + Tween 20 | Buffer | Fisher Scientific | BP-2471-1 | Original source unspecified |
| XCell II blot module | Instrument | Life Technologies | EI9051 | Original unspecified |
| Acetonitrile, HPLC grade | Chemical | Spectrum | HP412 | Original source unspecified |
| Chloroform | Chemical | Fisher | C606-4 | Original unspecified |
| D-alpha-hydroxyglutaric acid disodium salt (2HG) | Chemical | Santa Cruz Biotechnology | Sc-227739 | Replaces original from Sigma |
| Methanol, HPLC grade | Chemical | MP | 300141 | Original source unspecified |
| N-methyl-N-tert-butyldimethylsilyl trifluoroacetamide (MTBSTFA) | Chemical | Soltec Ventures | GC102 | Replaces original from Regis |
| Norvaline | Chemical | Sigma | N7627 | Original unspecified |
| Protein Concentration Assay; Quick Start Bradford Assay | Reporter assay | Bio-Rad | 500-0205 | Original unspecified |
| Lipofectamine 2000 | Cell culture | Life Technologies | 11668027 | Original cat # unspecified |

## Procedure

Note: 293T cells are maintained in DMEM with 10% FBS at 37°C/5% $CO_2$ All cells will be sent for STR profiling and mycoplasma testing.

- Using the pLPC backbone and the OriGene ORF clones, clone in the sequences for wild-type *IDH1*, wild-type *IDH2, IDH1*[R132H] or *IDH2*[R172K] to generate the following vectors:
    1. pLPC-*IDH1*
    2. pLPC-*IDH2*
    3. pLPC-*IDH1*[R132H]
    4. pLPC-*IDH2*[R172K]
- Prep each vector using an endo-free maxiprep kit according to the manufacturer's instructions.
- Confirm plasmid identity by sequencing insert and vector integrity by agarose gel electrophoresis.
    1. Note; OriGene ORF clones are shipped with sequencing primers.
- Plate 293T cells in #10 cm tissue culture dishes and let adhere overnight.
    1. Plate two plates; one will be harvested for Western blot (Step 3), the other for metabolite analysis (Step 4).
- Transfect 293T cells with appropriate plasmids with Lipofectamine 2000 according to the manufacturer's instructions.
    1. Note: Prepare separate transfection mixtures for each replicate, then use the same mixture for all plates within that replicate; do not use a single large volume for transfection mixture for all replicates.
        a. pLPC (empty vector)
        b. pLPC-*IDH1*
        c. pLPC-*IDH2*
        d. pLPC-*IDH1*[R132H]
        e. pLPC-*IDH2*[R172K]
    2. Incubate for 3 days.
        a. At this point, the matched plates for each replicate will be harvested; one plate for Western blot analysis (Step 6), the matched plate for GC-MS analysis (Step 7).
    3. Note; from this point forward, the analysis of each replicate must be conducted separately and independently from the other replicates. For example, each replicate should be run on its own gels.

- Western blot analysis of methylation status:
    1. Acid extraction of histones:
        a. Lyse cells in hypotonic lysis buffer for 1.
            i. Hypotonic lysis buffer: 10mM HEPES, 10mM KCl, 1.5mM $MgCl_2$, 0.5mM DTT, [#]protease inhibitor cocktail
        b. Add $H_2SO_4$ to 0.2N and incubate at 4°C overnight with rotation.
        c. [#]Centrifuge samples at 6,500x*g* for 10min at 4°C to pellet debris.
        d. Precipitate proteins with 33% TCA.
        e. Wash with acetone.
        f. Resuspend in de-ionized water.
    2. [*#]Quantify protein concentration using a Bradford Assay.
    3. [#]Load ~ 50 µg of protein per well and separate proteins on a 10% NuPAGE 4-12% gradient gel.
    4. [#]Transfer to nitrocellulose membrane using an XCell II blot module at 25 V for 1-2 hr (start at 100 mA per gel).
    5. *Perform a Ponceau stain and image to confirm transfer of proteins.
        a. Wash out Ponceau.
    6. Block membrane for [#]1 hr in 5% non-fat milk in PBS with 0.5% Tween-20.
    7. Incubate with primary antibodies [#]diluted in TBST + 1% nonfat milk at 4°C overnight. [*]Use the manufacturer's recommended dilution.
        a. Anti-IDH1
        b. Anti-IDH2
        c. Anti-H3K4me3
        d. Anti-H3K9me2
        e. Anti-H3K9me3
        f. Anti-H3K36me3
        g. Anti-H3K79me2
        h. Anti-H3 (loading control)
            i. [#]Each antibody will have its own gel run. Membranes will not be stripped and reprobed.
    8. Wash membrane [#]twice with TBST for a total of 20 min.
        a. Incubate with HRP-conjugated secondary antibodies [#]diluted in TBST for 1 hr at room temperature. [#*]Use manufacturer recommended dilutions.
    9. Wash three times with TBST.
    10. Detect signal [#]using ECL plus according to the manufacturer's instructions.
    11. Quantify band intensities with ImageJ.
        a. Normalize methylation band intensities to total H3.
        b. Divide normalized band intensities by the vector control band intensity.
- Gas chromatography-mass spectrometry analysis of 2HG levels. Note; the data in the original paper and the methodology are derived from Figueroa and colleagues (*Figueroa et al., 2010*).
    1. Gently remove culture medium from cells 3 days after transfection, [#]wash cells quickly three times with 2 ml ice-cold PBS, and add [#]0.5 ml ice-cold 80% methanol containing 20 µM L-norvaline per well of a 6-well plate to the cells.
        a. [#]Quantify protein concentration using the [#]Bio-Rad Quick Start Bradford Assay.
    2. Incubate 20 min at -80°C.
    3. Centrifuge at 14000x*g* for 20 min at 4°C.
        a. [#]Counter-extract samples with chloroform to remove nonpolar metabolites.
    4. Collect supernatant and dry using a [#]MiVac.
    5. Redissolve dried extracts in [#]60 µL of a 1:1 mixture of acetonitrile and N-methyl-N-tert-butyldimethylsilyltrifluoroacetamide (MTBSTFA).
    6. Heat the samples for 75 min at 70°C.
    7. GC-MS analysis:
        a. [#]A Shimadzu QP2010 Plus GC-MS is programmed with an injection temperature of 250°C, injection split ratio 1/10, with injection volume 0.3-1 µl. GC oven temperature starts at 130°C for 4 min, rising to 243°C at 6°C/min and to 280°C at 60°C/min with a final hold at this temperature for 2 min. GC flow rate with helium carrier gas was 50 cm/s. The GC column used is a 15 m x 0.25 mm x 0.25 µm Rxi-5ms (Restek). GC-MS interface temperature is 300°C and (electron impact) ion source temperature is 200°C, with 70 V/ 70 µA ionization voltage/ current. The mass spectrometer is set to scan m/z range 150-600, with ~1 kV detector sensitivity (modified as necessary).

8. *#In parallel to the sample, run a standard curve of known amounts of 2HG.
9. Confirm and *#quantify 2HG metabolite peak using standard curve.
10. Analyze and *#quantify 2HG and glutamate signal (identified by elution time and mass fragment pattern) intensities by integration of peak areas.
   - Repeat independently from Step 4 onwards an additional five times.

## Deliverables

- Data to be collected:
  - Sequence files and agarose gel images confirming vector identity
  - Full gel images of western blots with ladder (as seen in Figure 1B)
    - Images of Ponceau stained membranes
  - Quantification of band intensities (as seen in Supplemental Figure 1A)
  - GC-MS data
  - Quantification of signal intensities of 2HG and glutamate (as seen in Figure 1B)

## Confirmatory analysis plan

- Statistical Analysis of the Replication Data:
  Note: At the time of analysis we will calculate Pearson's *r* to check for correlation between the six dependent variables, normalized intensities measured for each of the histone lysine methylations, for the Western blot data. We will also perform the Shapiro-Wilk test and generate a quantile-quantile plot to assess the normality of the Western blot data and 2HG/glutamate ratios. We will also perform Levene's test to assess homoscedasticity. If the data appear skewed we will perform a log transformation in order to proceed with the proposed statistical analysis. If the log transformation does not result in similar variance across groups, we will perform the equivalent non-parametric test listed in Power Calculations for this protocol.
  - Western blot:
    - MANOVA (six dependent variables are the normalized intensities for each of the histone lysine methylations; four independent variables are the IDH1 and IDH2 variants (all normalized to vector) with the following planned comparisons using Bonferroni's correction:
      - Wild-type IDH1 compared to IDH1$^{R132H}$, for H3K9me2.
      - Wild-type IDH2 compared to IDH2$^{R172K}$, for H3K9me2.
  - 2HG/glutamate ratios:
    - One-way ANOVA (one dependent variable is the 2HG/glutamate ratio; four independent variables are the IDH1 and IDH2 variants) with the following planned comparisons using Bonferroni's correction:
      - IDH1$^{WT}$ compared to IDH1$^{R132H}$
      - IDH2$^{WT}$ compared to IDH2$^{R172K}$
- Meta-analysis of original and replication attempt effect sizes:
  - For Western blot:
    - The replication attempt will perform the statistical analysis listed above, compute the effects sizes, compare them against the reported effect size in the original paper and use a meta-analytic approach to combine the original and replication effects, which will be presented as a forest plot.
  - For 2HG/glutamate ratios:
    - The replication data will be presented as a mean with 95% confidence intervals and will include the original data point, calculated directly from the representative image, as a single point on the same plot for comparison.
- Additional exploratory analysis:
  - Correlation analysis (Pearson's *r*) of each of the six relative histone methylation levels to 2HG/glutamate levels using Bonferroni 's correction (as seen in Supplemental Figure 1B).

### Known differences from the original study

- The replication attempt will quantify total amounts of 2HG in addition to the ratio of 2HG to glutamate.
- Aspects of the Western blot protocol are provided by the replicating lab; complete details of the original protocol were unavailable.

### Provisions for quality control

All data obtained from the experiment - raw data, data analysis, control data and quality control data - will be made publicly available, either in the published manuscript or as an open access data-set available on the Open Science Framework (https://osf.io/vfsbo/).

- Sequence files and agarose gel images confirming vector identity and integrity
- Ponceau stains confirming protein transfer to membranes
- STR profiling and mycoplasma testing results

## Protocol 2: Examining the effects of mutations in IDH2 on differentiation of 3T3-L1 cells

This protocol describes how to induce the differentiation of 3T3-L1 cells into adipocytes, which involves extensive chromatin remodeling, after transfection with wild type and mutant forms of *IDH2* and assess the level of differentiation by Oil-Red-O staining, as seen in Figure 2A and B, and adipocyte marker expression, as seen in Figure 2D.

### Sampling

- This experiment will use 5 biological replicates for a minimum power of 80%. The metabolite data is qualitative, thus to determine an appropriate number of replicates to initially perform, sample sizes based on a range of potential variance was determined.
  - See Power Calculations for details.
- Each experiment will consist of three cohorts:
  - Cohort 1: 3T3-L1 cells transduced with vector
  - Cohort 2: 3T3-L1 cells transduced with wild-type *IDH2*
  - Cohort 3: 3T3-L1 cells transduced with *IDH2*$^{R172K}$
- Each cohort will have 5 plates per biological replicate:
  - One plate will be used to assess IDH2 expression by Western blot.
  - The second plate will be used to assess intracellular levels of 2HG.
  - The third plate will be assessed for adipogenesis by Oil-Red-O staining.
  - The fourth and fifth plates will have mRNA harvested for qRT-PCR analysis.

### Materials and reagents

| Reagent | Type | Manufacturer | Catalog # | Comments |
|---|---|---|---|---|
| Isobutylmethylxanthine | Inhibitor | Sigma | I5879 | Original cat# unspecified |
| Dexamethasone | Chemical | Sigma | D4902 | Original cat# unspecified |
| Insulin | Growth factor | Sigma | I3536 | Original cat# unspecified |
| Troglitazone | Chemical | Sigma | T2573 | Original cat# unspecified |
| pCL-Eco helper plasmid | Nucleic acid | Addgene | 12371 | Original source unspecified |
| 293T cells | Cell line | ATCC | CRL-3216 | Original source unspecified |
| 3T3-L1 cells | Cell line | ATCC | CL-173 | Original source unspecified |
| Puromycin | Chemical | Life Technologies | A11138-02 | Original unspecified |
| RIPA buffer | Cell culture | Millipore | 20188 | Original source unspecified |
| Nitrocellulose membrane | Western blot reagent | Life Technologies | LC2006 | Original source unspecified |

*Continued on next page*

*Continued*

| Reagent | Type | Manufacturer | Catalog # | Comments |
|---|---|---|---|---|
| Ponceau stain | Chemical | Sigma | P7170 | Original source unspecified |
| Rabbit polyclonal anti-IDH1 | Antibody | ProteinTech | 12332-1-AP | |
| Mouse IgG1 monoclonal anti-IDH2 | Antibody | Abcam | Ab55271 | |
| HRP-conjugated donkey anti-rabbit secondary | Antibody | GE Healthcare | NA934V | |
| HRP-conjugated sheep anti-mouse secondary | Antibody | GE Healthcare | NA931V | |
| Oil-Red-O | Chemical | Sigma | O1391 | Original source unspecified |
| paraformaldehyde | Chemical | Tousimis | 1008A | Original source unspecified |
| 6-well tissue culture plates | Labware | Sarstedt | 83.1839 | Original unspecified |
| XtremeGene HP reagent | Cell culture | Roche | 06366244001 | Original unspecified |
| DMEM | Cell culture | Corning | 15013 CV | Replaces original from Invitrogen |
| FBS | Cell culture | CellGro | 10437-028 | Original cat # unspecified |
| OPTI-MEM | Cell culture | Life Technologies | 31986070 | Original unspecified |
| 0.45 µm low binding syringe filter | Labware | Millipore | SLHV013SL | Original unspecified |
| Endo-free maxiprep kit | Kit | Qiagen | 12362 | Original unspecified |
| Protein Concentration Assay; Quick Start Bradford Assay | Reporter assay | Bio-Rad | 500-0205 | Original unspecified |
| Spectrophotometer | Instrument | Beckman Coulter | DU800 | Original unspecified |
| TRIzol | Chemical | Invitrogen | 15596-018 | Original cat# unspecified |
| SuperScript II reverse transcriptase | Kit | Invitrogen | 18064-014 | Original cat# unspecified |
| 7900HT Sequence Detection System | Instrument | Applied Biosystems | | |
| Pparg Taqman assays; Hs00234592_m1 | Nucleic acid | Applied Biosystems | Cat. # 4351372 | Original assay unspecified |
| Cebpa Taqman assays; Hs00269972_s1 | Nucleic acid | Applied Biosystems | Cat. # 4331182 | Original assay unspecified |
| Adipoq Taqman assays; Hs00605917_m1 | Nucleic acid | Applied Biosystems | Cat. # 4331182 | Original assay unspecified |
| 18S rRNA Taqman assays; Hs99999901_s1 | Nucleic acid | Applied Biosystems | Cat. # 4331182 | Original assay unspecified |

## Procedure

Note: 3T3-L1 and 293T cells are maintained in DMEM with 10% FBS at 37°C/5% $CO_2$. All cells will be sent for STR profiling and mycoplasma testing. pLPC (empty vector), pLPC-*IDH2*, and pLPC-*IDH2*[R172K] are generated in Protocol 1.

- Generate vector and *IDH2* wild type and mutant expressing retroviruses.
  1. [#]Transfect 293T cells with pCL-Eco helper plasmid and vector or *IDH2* vectors.
     a. Spot virus construct in 6 well plates at 1000 ng/well.
     b. Perform X-tremeGeneHP reverse transfection as follows:
        i. Make the helper plasmid mix; 700 ng/well.
        ii. Add 4 µL of XtremeGene HP to 400 µL OPTI-MEM.
           - Mix by light tapping.
        iii. Mix together the helper plasmids with the XtremeGeneHP reagent and OPTI-MEM.
        iv. Add 400 µL of the mix to each well and incubate for at least 30 min at room temperature.
        v. Meanwhile, resuspend 293T cells in DMEM + 10% FBS at $1.2 \times 10^6$ cells/ml.
        vi. Add 1600 µL of cells to each well.
  2. [#]24 hr later, replace media (2 ml total).
  3. [#]48 hr post transfection, collect supernatant from each well.
     a. Centrifuge at 500x*g* for 10 min at room temperature to pellet debris.
     b. Filter supernatant through a 0.45 µm syringe filter, aliquot and store at -80°C.
- Transduce 3T3-L1 cells with viral supernatant.
  1. [#]Seed cells in 6-well plates and incubate overnight.
     a. Cells should be 50-60% confluent the next day.

 2. #Add viral supernatant to medium.
- a. Supernatant will be added at varying concentrations to determine optimal transduction efficiency; 1:5 to 1:10 – 150 to 300 µL per well.

 3. #Adjust media volume to 1.4 ml per well.

 4. #Add polybrene in 100 µL of media into each well for a final concentration of 8 µg/ml.

 5. #Spinoculate by spinning at 1000x*g* for 60 min at room temperature.
- a. Incubate overnight.

 6. #Change media to remove viral transduction media.
- a. Replace with fresh media.

 7. Grow cells with 2.5 µg/ml puromycin for 7 days to select for stable expression of either wild-type or mutant *IDH2*.
- a. Maintain cells in puromycin.
- b. Also treat a non-transduced well of 3T3-L1 cells as a control showing susceptibility to puromycin.
- c. Split each biological replicate into 5 plates for the four assays being performed.
  - i. Plate 1 is for Western blot
  - ii. Plate 2 is for GC-MS
  - iii. Plate 3 is for Oil-Red-O staining (harvested 7 days after differentiation)
  - iv. Plate 4 and 5 are for qRT-PCR (harvested 0 and 4 days after differentiation)

- Generate whole cell lysates from the first plate of each cohort:
  1. Lyse cells and sonicate in RIPA buffer.
     - a. RIPA buffer: 1% sodium deoxycholate, 0.1% SDS, 1% Triton X-100, 0.01 M Tris pH 8.0 and 0.14 M NaCl
     - b. #Sonicate for 1 min, at 180 watts with rounds of 10 sec on/10 sec off. Keep sample on ice during sonication.
  2. Centrifuge lysates at 14000x*g* for 10 min at 4°C.
  3. Collect supernatant and measure total protein concentration #using a Bradford assay.
  4. Perform Western blot as outlined in Protocol 1 Step 6 using the following primary antibodies *at the manufacturer's recommended dilution:
     - a. Anti-IDH1
     - b. Anti-IDH2

- Harvest the second plate for metabolite analysis by mass spectrometry as described in Protocol 1 Step 4.

- Induce 3T3-L1 cells to differentiate into adipocytes.
  1. Incubate cells for 2 days with a differentiation cocktail composed of 0.5 mM isobutylmethylxanthine, 1 mM dexamethasone, 5 mg/ml insulin and 5 mM troglitazone supplementing the standard media.
  2. After 3 days, maintain cells with 5 mg/ml insulin until harvested.

- After 7 days of differentiation, assess adipogenesis by Oil-Red-O staining in the third plate from each cohort.
  1. Wash cells in PBS and fix in 3% paraformaldehyde for 20 min at room temperature.
  2. Wash cells with de-ionized water.
  3. Stain with Oil-Red-O solution according to the manufacturer's protocol.
  4. Image stained wells by brightfield microscopy and *quantify Oil-Red-O staining by extracting oil-red-o in isopropanol and reading absorbance at 500 nm.

- Harvest fourth plate for RNA extraction at Day 0 of differentiation and the fifth plate at Day 4 of differentiation and perform qRT-PCR to assess expression levels of adipocyte markers at each time point.
  1. Harvest cells and extract RNA using TRIzol according to the manufacturer's instructions.
  2. Reverse transcribe RNA and synthesize cDNA using SuperScript II reverse transcriptase according to the manufacturer's protocol.
     - a. Assess purity and concentration of RNA and cDNA spectrophotometrically; record $A_{260}/A_{280}$ and $A_{260}/A_{230}$ ratios.
  3. Perform qPCR on a 7900HT Sequence Detection system using Taqman gene expression assays for the following genes:
     - a. Pparg
     - b. Cebpa
     - c. Adipoq
     - d. 18S rRNA for normalization.
       - i. *Primers sequences and PCR cycling conditions will be optimized.

- Repeat independently from Step 2 onwards an additional four times.

## Deliverables

- Data to be collected:
  - Whole gel images of Western blots with ladder (as seen in Figure 2A)
  - *Densitometric quantification of bands
    - Also normalized to the loading control.
  - Images of wells stained with Oil-Red-O (as seen in Figure 2B)
  - *Quantification of Oil-Red-O levels for each cohort
  - All raw qRT-PCR data
  - Graph of gene expression over time for each of the three adipocyte markers (as seen in Figure 2D)

## Confirmatory analysis plan

- Statistical Analysis of the Replication Data:
  Note: At the time of analysis we will calculate Pearson's *r* to check for correlation between the three dependent variables, normalized gene expression for each of the adipocyte markers, for the qRT-PCR data. We will also perform the Shapiro-Wilk test and generate a quantile-quantile plot to assess the normality of the qRT-PCR data and 2HG/glutamate ratios. We will also perform Levene's test to assess homoscedasticity. If the data appears skewed we will perform a log transformation in order to proceed with the proposed statistical analysis. If this is not possible we will perform the equivalent non-parametric test listed in Power Calculations for this protocol.
  - Western Blot:
    - Confirmatory; no analysis performed
  - 2HG/glutamate ratio:
    - One-way ANOVA (one dependent variable is the 2HG/glutamate ratio; three independent variables are the vector and *IDH2* variants) with the following planned comparison using Fisher's LSD correction:
      - $IDH2^{R172K}$ compared to $IDH2^{WT}$
  - qRT-PCR:
    - One-way MANOVA (three dependent variables are the normalized gene expression of each of the adipocyte markers on day 4; three independent variables are the vector and IDH2 variants) with the following planned comparisons using Bonferroni's correction:
      - $IDH2^{R172}$ compared to vector for each gene (three comparisons total)
      - $IDH2^{R172K}$ compared to $IDH2^{WT}$ for each gene (three comparisons total)
- Meta-analysis of original and replication attempt effect sizes:
  - For qRT-PCR:
    - The replication attempt will perform the statistical analysis listed above, compute the effects sizes, compare them against the reported effect size in the original paper and use a meta-analytic approach to combine the original and replication effects, which will be presented as a forest plot.
  - For 2HG/glutamate ratios:
    - The replication data will be presented as a mean with 95% confidence intervals and will include the original data point, calculated directly from the representative image, as a single point on the same plot for comparison.
- Additional exploratory analysis:
  - Oil-Red-O staining:
    - One-way ANOVA (one dependent variable is the $A_{500}$ readings; three independent variables are the vector and *IDH2* variants) with the following planned comparison using Fisher's LSD correction:
      - $IDH2^{R172K}$ compared to $IDH2^{WT}$

## Known differences from the original study

- Aspects of the Western blot protocol are provided by the replicating lab; complete details of the original protocol were unavailable.
- Aspects of the viral production protocol are adapted from the replicating lab's in-house protocol.
  - Viral supernatant will be collected only at 48 hr post-transection and will not be combined with viral supernatant collected at 72 hr.
- In addition to imaging Oil-Red-O stained plates, the replication attempt will quantify the amount of Oil-Red-O staining spectrophotometrically.

## Provisions for quality control

All data obtained from the experiment - raw data, data analysis, control data and quality control data - will be made publicly available, either in the published manuscript or as an open access dataset available on the Open Science Framework (https://osf.io/vfsbo/).

- Sequence files and agarose gel images confirming vector identity and integrity
- Ponceau stains confirming protein transfer to membranes
- STR profiling and mycoplasma testing results
- Absorbance data for RNA and cDNA

## Protocol 3: Assessing the role of KDM4C on differentiation of 3T3-L1 cells

This protocol describes how to treat 3T3-L1 cells with an siRNA against the histone demethylase KDM4C, whose activity is inhibited by 2HG, and assess the effect of loss of KDM4C activity on methylation and differentiation, as seen in Figure 4D and Supplemental Figure 8.

## Sampling

- This experiment will be repeated 3 times for a minimum power of 80%. The Western blot data is qualitative, thus to determine an appropriate number of replicates to initially perform, sample sizes based on a range of potential variance was determined.
  - See Power calculations for details.
- Each experiment consists of five cohorts:
  - Cohort 1: 3T3-L1 cells treated with scramble control siRNAs
  - Cohort 2: 3T3-L1 cells treated with siRNAs #1 against KDM4C
  - Cohort 3: 3T3-L1 cells treated with siRNAs #2 against KDM4C
  - Cohort 4: 3T3-L1 cells treated with siRNAs #3 against KDM4C
  - Cohort 5: untreated 3T3-L1 cells [additional control]
- Each cohort is induced to differentiate, followed by:
  - Assessment of methylation by Western blot for:
    - Anti-KDM4C
    - Anti-H3K9me3
    - Anti-H3
    - Anti-$\beta$-actin
  - Assessment of differentiation by Oil-Red-O staining

## Materials and reagents

| Reagent | Type | Manufacturer | Catalog # | Comments |
| --- | --- | --- | --- | --- |
| 3T3-L1 cells | Cell line | ATCC | CL-173 | Original source unspecified |
| DMEM | Cell culture | Corning | 15013 CV | Replaces original from Invitrogen |
| FBS | Cell culture | CellGro | 10437-028 | Original cat # unspecified |

*Continued on next page*

*Continued*

| Reagent | Type | Manufacturer | Catalog # | Comments |
|---|---|---|---|---|
| KDM4C siRNA #1 | Nucleic acid | Synthesis left to the discretion of the replicating lab | | |
| KDM4C siRNA #2 | Nucleic acid | Synthesis left to the discretion of the replicating lab | | |
| KDM4C siRNA #3 | Nucleic acid | Synthesis left to the discretion of the replicating lab | | |
| Scrambled control siRNA | Nucleic acid | Dharmacon | D-001810-01-20 | |
| Lipofectamine RNAiMAX | Cell culture | Invitrogen | 13778-030 | Original cat# unspecified |
| isobutylmethylxanthine | Inhibitor | Sigma | I5879 | Original cat# unspecified |
| dexamethasone | Chemical | Sigma | D4902 | Original cat# unspecified |
| insulin | Growth factor | Sigma | I3536 | Original cat# unspecified |
| Troglitazone | Chemical | Sigma | T2573 | Original cat# unspecified |
| RIPA buffer | Cell culture | Millipore | 20188 | Original source unspecified |
| Nitrocellulose membrane | Western blot reagent | Life Technologies | LC2006 | Original source unspecified |
| Ponceau stain | Chemical | Sigma | P7170 | Original unspecified |
| Mouse IgG$_{2a}$ monoclonal anti-β-actin | Antibody | Sigma | A5316 | |
| Rabbit IgG monoclonal anti-H3 | Antibody | Cell Signaling Technology | 4499 | |
| Rabbit polyclonal anti-H3K9me3 | Antibody | Abcam | Ab8898 | |
| Rabbit polyclonal anti-KDM4C | Antibody | Abcam | Ab85454 | |
| Oil-Red-O | Chemical | Sigma | O1391 | Original source unspecified |
| paraformaldehyde | Chemical | Tousimis | 1008A | Original source unspecified |

## Procedure

Note: 3T3-L1 cells are maintained in DMEM with 10% FBS at 37°C/5% $CO_2$. All cells will be sent for STR profiling and mycoplasma testing.

- Transfect with 3T3-L1 cells with siRNAs against KDM4C:
  1. Plate out equal densities of single cell suspensions of 3T3-L1 cells in #6-well plates.
     a. *Optimize the number of cells to plate per well.
     b. Plate out two plates per siRNA pool (control vs. siKDM4C).
        i. One will be harvested on Day 3 of differentiation for Western blot analysis.
        ii. One will be used on Day 7 of differentiation for Oil-Red-O analysis.
  2. Transfect with the following siRNAs at a final concentration of 40 nM using Lipofectamine RNAiMAX according to the manufacturer's instructions.

| | | | |
|---|---|---|---|
| **1** | | **Sense** | **5'-GCUUGAAUCUCCCAAGAUATT-3'** |
| | | **Antisense** | **5'-UAUCUU GGGAGAUUCAAGCTT-3'** |
| 2 | | Sense | 5'-CAAAGUAUCUUGGAUCAAATT-3' |
| | | Antisense | 5'-UUUGAUCCAAGAUACUUUGCC-3' |
| 3 | | Sense | 5'-GAGGAGUU UCGGGAGUUCAACAAAU-3' |
| | | Antisense | 5'-AUUUGUUGAACUCCCGAA ACUCCUC-3' |

   a. Transfect control wells with a scrambled control siRNA.
   b. Also plate control wells with no transfection.
  3. Incubate for 3 days.
- Induce differentiation of control siRNA and antisense siRNA transduced 3T3-L1 cells as specified in Protocol 2 Step 6.
- 3 days after differentiation, harvest one plate from each treatment and prepare whole cell lysates as specified in Protocol 2 Step 7.
- Perform Western blot analysis on all whole cell lysates from Day 3 as described in Protocol 2 Step 7.
  1. Probe with:
     a. Anti-KDM4C

 b. Anti-H3K9me3
 c. Anti-H3
 d. Anti-$\beta$-actin
 2. Quantify band intensities with ImageJ.
 a. Normalize H3K9me3 band intensities to total H3.
 b. Normalize KDM4C band intensities to ß-actin.

- At Day 7 of differentiation, assess level of differentiation by Oil-Red-O staining as specified in Protocol 2 Step 8.
    1. Image wells and quantify Oil-Red-O expression.
- Repeat experiment an additional two times.

## Deliverables

- Data to be collected:
    - Whole gel images of all Western blots with ladder (as seen in the top of Figure 4D)
    - Images of Oil-Red-O stained wells (as seen in the bottom half of Figure 4D)
- Quantification of Oil-Red-O staining at Day 7 of differentiation (compare to Supplemental Figure 8B)

## Confirmatory analysis plan

- Statistical Analysis of the Replication Data:
  Note: At the time of analysis we will calculate Pearson's $r$ to check for correlation between the two dependent variables, normalized intensities measured for KDM4C and H3K9me3, for the Western blot data. We will also perform the Shapiro-Wilk test and generate a quantile-quantile plot to assess the normality of the Western blot and Oil-Red-O data. We will also perform Levene's test to assess homoscedasticity. If the data appears skewed we will perform a log transformation in order to proceed with the proposed statistical analysis. If this is not possible we will perform the equivalent non-parametric test listed in Power Calculations for this protocol.
    - Quantification of Oil-Red-O staining:
        - One way ANOVA (one dependent variable is the $A_{500}$ readings; four independent variables are the control and three KDM4C siRNA sequences) with the following planned comparisons using Bonferroni's correction:
            - Control siRNA compared to KDM4C siRNA #1
            - Control siRNA compared to KDM4C siRNA #2
            - Control siRNA compared to KDM4C siRNA #3 [additional exploratory analysis]
    - Western blot:
        - One-way MANOVA (two dependent variables are the normalized intensities measured for KDM4C and H3K9me3; four independent variables are the control and three KDM4C siRNA sequences) with the following planned comparisons using Bonferroni's correction:
            - H3K9me3 levels:
                - Control siRNA compared to KDM4C siRNA #1
                - Control siRNA compared to KDM4C siRNA #2
                - Control siRNA compared to KDM4C siRNA #3
            - KDM4C levels (QC):
                - Control siRNA compared to KDM4C siRNA #1
                - Control siRNA compared to KDM4C siRNA #2
                - Control siRNA compared to KDM4C siRNA #3
- Meta-analysis of original and replication attempt effect sizes:
    - Oil-Red-O staining for siRNA #1 and #2:
        - This replication attempt will perform the statistical analysis listed above, compute the effects sizes, compare them against the reported effect size in the original paper and use a meta-analytic approach to combine the original and replication effects, which will be presented as a forest plot.
            - There is no originally reported data from siRNA #3, therefore it will not be included.
    - Western Blot:

- The replication data will be presented as a mean with 95% confidence intervals and will include the original data point, calculated directly from the representative image, as a single point on the same plot for comparison.

## Known differences from the original study

The replication will perform the Oil-Red-O quantification for all three siRNAs, not just #1 and #2 as presented in Supplemental Figure 8.

## Provisions for quality control

All data obtained from the experiment - raw data, data analysis, control data and quality control data - will be made publicly available, either in the published manuscript or as an open access data-set available on the Open Science Framework (https://osf.io/vfsbo/).

- Ponceau stains confirming protein transfer to membranes
- STR profiling and mycoplasma testing results

## Power calculations

Note: details of all power calculations can be found at https://osf.io/rb32p/

## **Protocol 1**

### Summary of original data

Note: data estimated from published figures.

| Supplemental Figure 1: normalized WB band intensity (normalized to Vector) | | Mean | SD | N |
|---|---|---|---|---|
| IDH1$^{WT}$ | H3K9me2 | 1.7 | 0.8 | 3 |
| | H3K9me3 | 1 | 0.2 | 3 |
| | K3K4me3 | 1.2 | 0.6 | 3 |
| | H3K27me3 | 0.4 | 0.3 | 3 |
| | H3K36me3 | 1.2 | 0.4 | 3 |
| | H3K27me2 | 0.8 | 0.4 | 3 |
| IDH1$^{R132H}$ | H3K9me2 | 7.9 | 2.5 | 3 |
| | H3K9me3 | 4.1 | 1.2 | 3 |
| | K3K4me3 | 3.4 | 0.8 | 3 |
| | H3K27me3 | 2.5 | 0.5 | 3 |
| | H3K36me3 | 1.7 | 0.8 | 3 |
| | H3K27me2 | 4.7 | 2.5 | 3 |
| IDH2$^{WT}$ | H3K9me2 | 3.2 | 1.1 | 3 |
| | H3K9me3 | 2.1 | 1.2 | 3 |
| | K3K4me3 | 1.9 | 0.3 | 3 |
| | H3K27me3 | 1.9 | 0.8 | 3 |
| | H3K36me3 | 1.4 | 0.4 | 3 |
| | H3K27me2 | 1.5 | 0.9 | 3 |
| IDH2$^{R172K}$ | H3K9me2 | 11.4 | 3.8 | 3 |
| | H3K9me3 | 4.9 | 1.6 | 3 |
| | K3K4me3 | 4 | 1.4 | 3 |
| | H3K27me3 | 3.6 | 1.6 | 3 |
| | H3K36me3 | 1.8 | 0.7 | 3 |
| | H3K27me2 | 5.4 | 3.7 | 3 |

**Figure 1B: 2HG/glutamate ratios**

|  | Mean | Assumed N |
|---|---|---|
| IDH1[WT] | 0.005 | 3 |
| IDH1[R132H] | 0.052 | 3 |
| IDH2[WT] | 0.023 | 3 |
| IDH2[R172K] | 1.56 | 3 |

## Test family

- Western blot; Figure 1B/Supplemental Figure 1A:
  Note: Since we do not have the raw data, we were unable to perform power calculations using a MANOVA. We are approximating sample sizes with corrected one-way ANOVAs for each DV (normalized histone methylations).
  - Bonferroni-corrected one-way ANOVAs (one per DV) followed by Bonferroni corrected planned comparisons:
    - Wild-type IDH1 compared to IDH1[R132H], collapsed across all histone lysine methylations.
    - Wild-type IDH2 compared to IDH2[R172K], collapsed across all histone lysine methylations.
      - Note: Only H3K9me2 is being included since this is the histone modification with the largest effect size reported. A correlation among all the histone methylations will also be performed prior to performing the proposed analysis plan.
- 2HG/glutamate ratios; Figure 1B:
  - One-way ANOVA followed by Bonferroni-corrected pairwise comparisons for the following:
    - IDH1[WT] compared to IDH1[R132H], for H3K9me2
    - IDH2[WT] compared to IDH2[R172K], for H3K9me2

## Power calculations

- Power calculations were performed using R software (version 3.2.2) (R *Core Team, 2015*) and G*Power (version 3.1.7) (*Faul et al., 2007*)
- Partial $\eta^2$ calculated as in *Lakens (2013)*
- Western blot calculations:
  - Note: Due to the large variance, these parametric tests are only used for comparison purposes. The sample size is based on the non-parametric tests also listed. For the ANOVA a Kruskal-Wallis would be performed as the non-parametric alternative, which would require an ~15% increase in sample size calculated for the parametric test listed.

**One-way ANOVA: α=0.00833, 4 groupsα**

| DV | F(3,8) | Partial η2 | Effect size *f* | A priori power | Total Sample Size |
|---|---|---|---|---|---|
| H3K9me2 | 10.486 | 0.79726 | 1.98302 | 92.1%[1] | 12[1] |
| H3K9me3 | 7.0274 | 0.72492 | 1.62335 | 96.4%[1] | 16[1] |
| H3K4me3 | 6.6197 | 0.71284 | 1.57556 | 95.1%[1] | 16[1] |
| H3K27me3 | 6.0339 | 0.69351 | 1.50423 | 92.7%[1] | 16[1] |
| H3K36me3 | 0.6276 | 0.19051 | 1.06125 | 90.2% | 24 |
| H3K79me2 | 3.0033 | 0.52969 | 1.07984[2] | 80.0%[2] | 24 |

[1] With 6 samples per group (24 total), achieved power is 99.9%.

[2] Since the original effect size will not be detectable with the proposed sample size, this is the effect size that can be detected at 80% power with the given sample size. The original effect size was 0.48512.

**Planned contrasts; two-tailed *t*-test: α=0.004167**

| Group 1 | Group 2 | Effect size *d* | A priori power | n/group |
|---|---|---|---|---|
| IDH1$^{WT}$ | IDH1$^{R132H}$ | 3.34039 | 84.9%[1] | 5[1] |
| IDH2$^{WT}$ | IDH2$^{R172K}$ | 2.93138 | 87.3% | 6 |

Planned contrasts; two-tailed Wilcoxon-Mann-Whitney: α=0.025

| Group 1 | Group 2 | Effect size *d* | A priori power | n/group |
|---|---|---|---|---|
| IDH1$^{WT}$ | IDH1$^{R132H}$ | 3.34039 | 80.9%[2] | 5[2] |
| IDH2$^{WT}$ | IDH2$^{R172K}$ | 2.93138 | 84.0% | 6 |

[1] With 6 samples per group, achieved power is 95.3%.
[2] With 6 samples per group, achieved power is 93.4%.

- 2HG/glutamate ratio calculations:
  - Note: The original data does not indicate the error associated with multiple biological replicates. To identify a suitable sample size, power calculations were performed using different levels of relative variance.

**2%; one-way ANOVA: α=0.05, 4 groups**

| $F_{(3,8)}$ | Partial η$^2$ | Effect size *f* | A priori power | Total Sample Size |
|---|---|---|---|---|
| 7240.7 | 0.99963 | 52.11901 | 99.9% | 8 |

Planned comparisons; 2-tailed *t*-test: α=0.025

| Group 1 versus | Group 2 | Effect size *d* | A priori power | n per group |
|---|---|---|---|---|
| IDH1$^{WT}$ | IDH1$^{R132H}$ | 63.6182 | 99.9% | 2 |
| IDH2$^{WT}$ | IDH2$^{R172H}$ | 69.6606 | 99.9% | 2 |

**15%; one-way ANOVA: α=0.05, 4 groups**

| $F_{(3,8)}$ | Partial η$^2$ | Effect size *f* | A priori power | Total Sample Size |
|---|---|---|---|---|
| 128.72 | 0.97970 | 6.94772 | 99.9% | 8 |

Planned comparisons; 2-tailed *t*-test: α=0.025

| Group 1 versus | Group 2 | Effect size *d* | A priori power | n per group |
|---|---|---|---|---|
| IDH1$^{WT}$ | IDH1$^{R132H}$ | 8.48242 | 83.5% | 2 |
| IDH2$^{WT}$ | IDH2$^{R172H}$ | 9.28808 | 88.4% | 2 |

**28%; one-way ANOVA: α=0.05, 4 groups**

| $F_{(3,8)}$ | Partial η2 | Effect size *f* | A priori power | Total Sample Size |
|---|---|---|---|---|
| 36.942 | 0.93268 | 3.72201 | 99.9% | 8 |

Planned comparisons; 2-tailed *t*-test: α=0.025

| Group 1 versus | Group 2 | Effect size *d* | A priori power | n per group |
|---|---|---|---|---|
| IDH1$^{WT}$ | IDH1$^{R132H}$ | 4.54415 | 92.4% | 3 |
| IDH2$^{WT}$ | IDH2$^{R172H}$ | 4.97576 | 95.9% | 3 |

**40%; one-way ANOVA: α=0.05, 4 groups**

| F(3,8) | Partial η2 | Effect size f | A priori power | Total Sample Size |
|---|---|---|---|---|
| 18.102 | 0.87160 | 2.60542 | 96.0% | 8 |

Planned comparisons; 2-tailed *t*-test: α=0.025

| Group 1 versus | Group 2 | Effect size *d* | A priori power | n per group |
|---|---|---|---|---|
| IDH1<sup>WT</sup> | IDH1<sup>R132H</sup> | 3.18091 | 89.6% | 4 |
| IDH2<sup>WT</sup> | IDH2<sup>R172H</sup> | 3.48303 | 94.2% | 4 |

In order to produce quantitative replication data, we will run the experiment six times. Each time we will quantify the 2HG/glutamate ratio. We will determine the standard deviation across the biological replicates and combine this with the reported value from the original study to simulate the original effect size. We will use this simulated effect size to determine the number of replicates necessary to reach a power of at least 80%. We will then perform additional replicates, if required, to ensure that the experiment has more than 80% power to detect the original effect.

## Protocol 2
Summary of original data
Note: data estimated from published figures.

| Figure 2A: 2HG/glutamate ratio | | Assumed N |
|---|---|---|
| | Mean | |
| Vector | 0.1 | 3 |
| IDH1<sup>R172K</sup> | 5.3 | 3 |
| IDH2<sup>WT</sup> | 0.1 | 3 |

| Figure 2D: Relative expression of adipocyte markers | | | | |
|---|---|---|---|---|
| Pparg | | Mean | SD | N |
| Vector | Day 0 | 1.45 | 0.823 | 3 |
| | Day 4 | 13.992 | 0.816 | 3 |
| IDH2<sup>WT</sup> | Day 0 | 2.521 | 1.076 | 3 |
| | Day 4 | 10.966 | 0.879 | 3 |
| IDH2<sup>R172K</sup> | Day 0 | 1.134 | 0.823 | 3 |
| | Day 4 | 4.223 | 0.941 | 3 |
| Cebpa | | Mean | SD | N |
| Vector | Day 0 | 1.123 | 0.421 | 3 |
| | Day 4 | 3.053 | 1.188 | 3 |
| IDH2<sup>WT</sup> | Day 0 | 1.93 | 0.456 | 3 |
| | Day 4 | 4.807 | 0.565 | 3 |
| IDH2<sup>R172K</sup> | Day 0 | 0.667 | 0.491 | 3 |
| | Day 4 | 0.246 | 0.21 | 3 |
| Adipoq | | Mean | SD | N |

*Continued on next page*

| Vector | Day 0 | 0 | 58.621 | 3 |
| | Day 4 | 572.414 | 193.103 | 3 |
| *IDH2*<sup>WT</sup> | Day 0 | 0 | 58.621 | 3 |
| | Day 4 | 448.276 | 86.207 | 3 |
| *IDH2*<sup>R172K</sup> | Day 0 | 0 | 58.621 | 3 |
| | Day 4 | 41.379 | 27.586 | 3 |

## Test family

- 2HG/glutamate ratios; Figure 2A bottom:
  - One way ANOVA followed by Fisher's LSD for the following comparison:
    - IDH2<sup>WT</sup> vs. IDH2<sup>R172K</sup>
- qRT-PCR; Figure 2D:
  Note: Since we did not have the raw data, we were unable to perform power calculations using a MANOVA. We are approximating the sample sizes with corrected one-way ANOVAs for each DV (gene).
  - Bonferroni-corrected one-way ANOVAs (one per gene) followed by Bonferroni corrected comparisons for Day 4 timepoints:
    - *IDH2*<sup>R172K</sup>compared to vector for each gene (3 comparisons total)
    - *IDH2*<sup>R172K</sup>compared to *IDH2*<sup>WT</sup> for each gene (3 comparisons total)

## Power calculations

- Power calculations were performed using R software (version 3.2.2) (R *Core Team, 2015*) and G*Power (version 3.1.7) (*Faul et al., 2007*)
- Partial $\eta^2$ calculated as in *Lakens (2013)*
- 2HG/glutamate ratios:
  - Note: The original data does not indicate the error associated with multiple biological replicates. To identify a suitable sample size, power calculations were performed using different levels of relative variance.

**2%; one-way ANOVA: α=0.05, 3 groups**

| F(2,6) | Partial η$^2$ | Effect size $f$ | A priori power | Total Sample Size |
| --- | --- | --- | --- | --- |
| 7214.5 | 0.99958 | 49.0188 | 99.9% | 6 |

Planned comparisons; two-tailed *t*-test: α=0.05

| Group 1 versus | Group 2 | Effect size $d$ | A priori power | n per group |
| --- | --- | --- | --- | --- |
| IDH2<sup>R172H</sup> | IDH2<sup>WT</sup> | 84.9383 | 99.9% | 2 |

**15%; one-way ANOVA: α=0.05, 3 groups**

| F(2,6) | Partial η2 | Effect size $f$ | A priori power | Total Sample Size |
| --- | --- | --- | --- | --- |
| 128.26 | 0.97715 | 6.53866 | 99.9% | 6 |

Planned comparisons; two-tailed *t*-test: α=0.05

| Group 1 versus | Group 2 | Effect size $d$ | A priori power | n per group |
| --- | --- | --- | --- | --- |
| IDH2<sup>R172H</sup> | IDH2<sup>WT</sup> | 11.3252 | 99.8% | 2 |

**28%; one-way ANOVA: α=0.05, 3 groups**

| F(2,6) | Partial η2 | Effect size $f$ | A priori power | Total Sample Size |
| --- | --- | --- | --- | --- |

| 36.809 | 0.92464 | 3.50285 | 98.5% | 6 |

Planned comparisons; two-tailed *t*-test: α=0.05

| Group 1 versus | Group 2 | Effect size *d* | A priori power | n per group |
|---|---|---|---|---|
| IDH2^R172H | IDH2^WT | 6.06704 | 84.2% | 2 |

**40%; one-way ANOVA: α=0.05, 3 groups**

| F(2,6) | Partial η2 | Effect size *f* | A priori power | Total Sample Size |
|---|---|---|---|---|
| 18.036 | 0.85739 | 2.45194 | 85.1% | 6 |

Planned comparisons; two-tailed *t*-test: α=0.05

| Group 1 versus | Group 2 | Effect size *d* | A priori power | n per group |
|---|---|---|---|---|
| IDH2^R172H | IDH2^WT | 4.24688 | 96.6% | 3 |

In order to produce quantitative replication data, we will run the experiment five times. Each time we will quantify the 2HG/glutamate ratio. We will determine the standard deviation across the biological replicates and combine this with the reported value from the original study to simulate the original effect size. We will use this simulated effect size to determine the number of replicates necessary to reach a power of at least 80%. We will then perform additional replicates, if required, to ensure that the experiment has more than 80% power to detect the original effect.

- qRT-PCR:
  - Note: Due to the large variance, these parametric tests are only used for comparison purposes. The sample size is based on the non-parametric tests also listed. For the ANOVA a Kruskal-Wallis would be performed as the non-parametric alternative, which would require an ~15% increase in sample size calculated for the parametric test listed.

**Pparg**

One-way ANOVA: α=0.0167, 3 groups

| F(2,6) | Partial η$^2$ | Effect size *f* | A priori power | Total Sample Size |
|---|---|---|---|---|
| 96.854 | 0.96996 | 5.68195 | 99.5%[1] | 6[1] |

Planned comparisons; two-tailed *t*-test: α=0.0083

| Group 1 versus | Group 2 | Effect size *d* | A priori power | n per group |
|---|---|---|---|---|
| IDH2^R172H | Vector | 11.0921 | 99.9%[2] | 3[2] |
| IDH2^R172H | IDH2^WT | 7.40559 | 98.6%[2] | 3[2] |

Planned comparisons; two-tailed Wilcoxon-Mann-Whitney: α=0.0083

| Group 1 versus | Group 2 | Effect size *d* | A priori power | n per group |
|---|---|---|---|---|
| IDH2^R172H | Vector | 11.0921 | 99.9%[2] | 3[2] |
| IDH2^R172H | IDH2^WT | 7.40559 | 96.9%[2] | 3[2] |

[1] With 5 samples per group (15 total), achieved power is 99.9%.
[2] With 5 samples per group, achieved power is 99.9%.

**Cebpa**

One-way ANOVA: α=0.0167, 3 groups

| F(2,6) | Partial η$^2$ | Effect size *f* | A priori power | Total Sample Size |
|---|---|---|---|---|
| 26.843 | 0.89947 | 5.68195 | 90.5%[1] | 6[1] |

Planned comparisons; two-tailed *t*-test: α=0.0083

| Group 1 versus | Group 2 | Effect size | A priori power | n per group |
|---|---|---|---|---|
| $IDH2^{R172H}$ | Vector | 3.29048 | 91.6% | 5 |
| $IDH2^{R172H}$ | $IDH2^{WT}$ | 10.7011 | 99.9%[2] | 3[2] |

Planned comparisons; two-tailed Wilcoxon-Mann-Whitney: $\alpha$=0.0083

| Group 1 versus | Group 2 | Effect size d | A priori power | n per group |
|---|---|---|---|---|
| $IDH2^{R172H}$ | Vector | 3.29048 | 89.0% | 5 |
| $IDH2^{R172H}$ | $IDH2^{WT}$ | 10.7011 | 99.9%[2] | 3[2] |

[1] With 5 samples per group (15 total), achieved power is 99.9%.
[2] With 5 samples per group, achieved power is 99.9%.

**Adipoq**

One-way ANOVA: $\alpha$=0.0167, 3 groups

| F(2,6) | Partial $\eta^2$ | Effect size f | A priori power | Total Sample Size |
|---|---|---|---|---|
| 15.269 | 0.83579 | 2.25603 | 96.3%[1] | 9[1] |

Planned comparisons; two-tailed t-test: $\alpha$=0.0083

| Group 1 versus | Group 2 | Effect size | A priori power | n per group |
|---|---|---|---|---|
| $IDH2^{R172H}$ | Vector | 3.85001 | 87.8%[2] | 4[2] |
| $IDH2^{R172H}$ | $IDH2^{WT}$ | 6.35752 | 94.5%[3] | 3[3] |

Planned comparisons; two-tailed Wilcoxon-Mann-Whitney: $\alpha$=0.0083

| Group 1 versus | Group 2 | Effect size d | A priori power | n per group |
|---|---|---|---|---|
| $IDH2^{R172H}$ | Vector | 3.85001 | 84.0%[4] | 4[4] |
| $IDH2^{R172H}$ | $IDH2^{WT}$ | 6.35752 | 90.8%[3] | 3[3] |

[1] With 5 samples per group (15 total), achieved power is
99.9%.
[2] With 5 samples per group, achieved power is 97.9%.
[3] With 5 samples per group, achieved power is 99.9%.
[4] With 5 samples per group, achieved power is 96.8%.

## Protocol 3
## Summary of original data
Note: data estimated from published figures.

| **Figure 4D and S8A: Western Blot** | | **Band intensity (normalized to H3)** |
|---|---|---|
| KDM4C | Control siRNA | 1 |
| | KDM4C siRNA #1 | 0.5097[1] |
| | KDM4C siRNA #2 | 0.2767[1] |
| | KDM4C siRNA #3 | 0.0249[2] |
| H3K9me3 | Control siRNA | 1 |
| | KDM4C siRNA | 0.3695[2] |

[1] These values were normalized to ß-Actin as seen in Supplemental Figure 8A.
[2] These values were normalized to total H3 as seen in Figure 4D. Also there is no data for siRNAs #1 and #2. We have assumed similar values for siRNA #3 for the purposes of these calculations.

| Supplemental Figure 8B: Oil-Red-O quantification | Mean | SD | N |
|---|---|---|---|
| Control | 1.06 | 0.03 | 3 |
| siRNA #1 | 0.42 | 0.15 | 3 |
| siRNA #2 | 0.69 | 0.09 | 3 |
| siRNA #3[1] | 0.69 | 0.09 | 3 |

[1] There is no data for siRNA #3. We have assumed similar values as siRNA #2 for the purposes of these calculations.

## Test family

- Western blot; Figure 4D and S8A:
  Note: Since we did not have the raw data, we were unable to perform power calculations using a MANOVA. We are approximating the sample sizes with corrected one-way ANOVAs for each DV (normalized protein).
  - One-way ANOVAs followed by Bonferroni corrected comparisons:
    - H3K9me3 levels in control siRNA compared to each *KDM4C* siRNA (3 comparisons total)
    - *KDM4C* levels in control siRNA compared to each KDM4C siRNA (3 comparisons total)
- Quantification of Oil-Red-O staining; Figure S8B:
  - One way ANOVA followed by Bonferroni corrected comparisons:
    - Control siRNA compared to *KDM4C* siRNA #1
    - Control siRNA compared to *KDM4C* siRNA #2
    - Control siRNA compared to *KDM4C* siRNA #3

## Power calculations

- Power calculations were performed using R software (version 3.2.2) (R *Core Team, 2015*) and G*Power (version 3.1.7) (*Faul et al., 2007*).
- Partial $\eta^2$ calculated as in *Lakens (2013)*.
- Western Blot
  - Note: The original data does not indicate the error associated with multiple biological replicates. To identify a suitable sample size, power calculations were performed using different levels of relative variance.

**2%; One-way ANOVA: α=0.025, 4 groups α**

| DV | F(3,8) | Partial $\eta^2$ | Effect size $f$ | Power | Total sample size |
|---|---|---|---|---|---|
| H3K9me3 | 2114.7 | 0.99874 | 28.161 | 99.9% | 8 |
| *KDM4C* | 3865.5 | 0.99931 | 38.073 | 99.9% | 8 |

Planned comparisons; two-tailed *t*-test: α=0.0083

| DV | Group 1 versus | Group 2 | Effect size $d$ | A priori power | n per group |
|---|---|---|---|---|---|
| H3K9me3 | Control | KDM4C #1 | 53.099 | 99.9% | 2 |
| | Control | KDM4C #2 | 53.099 | 99.9% | 2 |
| | Control | KDM4C #3 | 53.099 | 99.9% | 2 |
| KDM4C | Control | KDM4C #1 | 42.407 | 99.9% | 2 |
| | Control | KDM4C #2 | 62.552 | 99.9% | 2 |
| | Control | KDM4C #3 | 84.335 | 99.9% | 2 |

**15%; One-way ANOVA: α=0.025, 4 groups**

| DV | F(3,8) | Partial $\eta^2$ | Effect size $f$ | Power | Total sample size |
|---|---|---|---|---|---|
| H3K9me3 | 37.595 | 0.93377 | 3.7547 | 99.2% | 8 |
| KDM4C | 68.72 | 0.96264 | 5.0764 | 99.9% | 8 |

Planned comparisons; two-tailed $t$-test: α=0.0083

| DV | Group 1 versus | Group 2 | Effect size $d$ | A priori power | n per group |
|---|---|---|---|---|---|
| H3K9me3 | Control | KDM4C #1 | 7.0800 | 97.8% | 3 |
| | Control | KDM4C #2 | 7.0800 | 97.8% | 3 |
| | Control | KDM4C #3 | 7.0800 | 97.8% | 3 |
| KDM4C | Control | KDM4C #1 | 5.6543 | 88.5% | 3 |
| | Control | KDM4C #2 | 8.3403 | 99.6% | 3 |
| | Control | KDM4C #3 | 11.245 | 99.9% | 3 |

**28%; One-way ANOVA: α=0.025, 4 groups**

| DV | F(3,8) | Partial $\eta^2$ | Effect size $f$ | Power | Total sample size |
|---|---|---|---|---|---|
| H3K9me3 | 10.789 | 0.80182 | 2.0114 | 98.7% | 12 |
| KDM4C | 19.722 | 0.88089 | 2.7195 | 89.2% | 8 |

Planned comparisons; two-tailed $t$-test: α=0.0083

| DV | Group 1 versus | Group 2 | Effect size $d$ | A priori power | n per group |
|---|---|---|---|---|---|
| H3K9me3 | Control | KDM4C #1 | 3.7928 | 86.8% | 4 |
| | Control | KDM4C #2 | 3.7928 | 86.8% | 4 |
| | Control | KDM4C #3 | 3.7928 | 86.8% | 4 |
| KDM4C | Control | KDM4C #1 | 3.0291 | 85.9% | 5 |
| | Control | KDM4C #2 | 4.4680 | 95.8% | 4 |
| | Control | KDM4C #3 | 6.0240 | 92.1% | 3 |

**40%; One-way ANOVA: α=0.025, 4 groups**

| DV | F(3,8) | Partial $\eta^2$ | Effect size $f$ | Power | Total sample size |
|---|---|---|---|---|---|
| H3K9me3 | 5.2868 | 0.66472 | 1.4080 | 80.2% | 12 |
| KDM4C | 9.6637 | 0.78373 | 1.9037 | 97.6% | 12 |

Planned comparisons; two-tailed $t$-test: α=0.0083

| DV | Group 1 versus | Group 2 | Effect size $d$ | A priori power | n per group |
|---|---|---|---|---|---|
| H3K9me3 | Control | KDM4C #1 | 2.6550 | 87.2% | 6 |
| | Control | KDM4C #2 | 2.6550 | 87.2% | 6 |
| | Control | KDM4C #3 | 2.6550 | 87.2% | 6 |
| KDM4C | Control | KDM4C #1 | 2.1204 | 85.6% | 8 |
| | Control | KDM4C #2 | 3.1276 | 88.3% | 5 |
| | Control | KDM4C #3 | 4.2168 | 93.3% | 4 |

In order to produce quantitative replication data, we will run the experiment three times. Each time we will quantify band intensity. We will determine the standard deviation of band intensity across the biological replicates and combine this with the reported value from the original study to simulate the original effect size. We will use this simulated effect size to determine the number of replicates necessary to reach a power of at least 80%. We will then perform additional replicates, if required, to ensure that the experiment has more than 80% power to detect the original effect.

- Oil-Red-O staining:
  - Note: Due to the large variance, these parametric tests are only used for comparison purposes. The sample size is based on the non-parametric tests also listed. For the ANOVA a Kruskal-Wallis would be performed as the non-parametric alternative, which would require an ~15% increase in sample size calculated for the parametric test listed.

**One-way ANOVA: α=0.05, 4 groups**

| F(3,8) | Partial eta$^2$ | Effect size $f$ | Power | Total Sample Size |
|---|---|---|---|---|
| 20.939 | 0.88703 | 2.8022 | 97.8%[1] | 8[1] |

[1] With 3 samples per group (12 total), achieved power is 99.9%.

**Planned comparisons; two-tailed Wilcoxon-Mann-Whitney: α=0.0167**

Power Calculations

| Group 1 | Group 2 | Effect size $d$ | Power | n/group |
|---|---|---|---|---|
| Control | KDM4C #1 | 5.9168 | 96.9% | 3 |
| Control | KDM4C #2 | 5.5156 | 93.3% | 3 |
| Control | KDM4C #3 | 5.5156 | 93.3% | 3 |

Planned comparisons; two-tailed $t$-test: α=0.0167

| Group 1 | Group 2 | Effect size $d$ | Power | n/group |
|---|---|---|---|---|
| Control | KDM4C #1 | 5.9168 | 97.7% | 3 |
| Control | KDM4C #2 | 5.5156 | 95.6% | 3 |
| Control | KDM4C #3 | 5.5156 | 95.6% | 3 |

# Acknowledgements

We thank Courtney Soderberg at the Center for Open Science for assistance with statistical analyses. We would also like to thanks the following companies for generously donating reagents to the Reproducibility Project: Cancer Biology; American Type and Tissue Collection (ATCC), Applied Biological Materials, BioLegend, Charles River Laboratories, Corning Incorporated, DDC Medical, EMD Millipore, Harlan Laboratories, LI-COR Biosciences, Mirus Bio, Novus Biologicals, Sigma-Aldrich, and System Biosciences (SBI).

# Additional information

## Group author details

Reproducibility Project: Cancer Biology

Elizabeth Iorns: Science Exchange, Palo Alto, United States; William Gunn: Mendeley, London, United Kingdom; Fraser Tan: Science Exchange, Palo Alto, United States; Joelle Lomax: Science Exchange, Palo Alto, United States; Nicole Perfito: Science Exchange, Palo Alto, United States; Timothy Errington: Center for Open Science, Charlottesville, United States

## Competing interests

ADR, DAS, OZ: Cancer Metabolism Facility at Sanford Burnham Prebys Medical Discovery Institute is a Science Exchange associated laboratory PA-B, C-CC: Functional Genomics Core at Sanford Burnham Prebys Medical Discovery Institute is a Science Exchange associated laboratory RP:CB: EI, FT,

JL, NP: Employed by and hold shares in Science Exchange Inc The other authors declare that no competing interests exist.

## Funding

| Funder | Author |
|---|---|
| Laura and John Arnold Foundation | The Reproducibility Project: Cancer Biology |

The Reproducibility Project: Cancer Biology is funded by the Laura and John Arnold Foundation, provided to the Center for Open Science in collaboration with Science Exchange.

## Author contributions

ADR, DAS, OZ, PA-B, C-CC, DAR-G, Drafting or revising the article; RP:CB, Conception and design, Drafting or revising the article

## Author ORCIDs

David A Russler-Germain, http://orcid.org/0000-0003-1009-2247

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
