## [Decision Letter]

Thank you for submitting your work entitled "Registered report: IDH mutation impairs histone demethylation and results in a block to cell differentiation" for consideration by *eLife*. Your article has been reviewed by three peer reviewers, and the evaluation has been overseen by Irwin Davidson (Reviewing Editor) and Randy Schekman (Senior Editor).

The reviewers have discussed the reviews with one another and the Reviewing editor has drafted this decision to help you prepare a revised submission.

Summary:

The reviewers found this proposed study to have important impact on the functional consequences of the frequently occurring IDH mutations and their mode of action. The provided protocol appears complete, but several points concerning the statistical methods remain to be clarified.

Essential revisions:

Normality and homoscedasticity will be assessed using adequate tests and methods. However what kind of transformation do the authors plan to apply in case of skewed data? Do the authors have an idea of how the data could be skewed? Similarly, which test do the authors intend to apply in case of non-normal data?

Power calculations assume normally distributed data. In the opposite case non parametric tests will be used with much less power. Power calculations should be taken with caution, as they are performed according to an optimistic scenario.

Protocol 1:

The authors should clarify if the MANOVA analyses are planned to analyze the results of western blots and that:

a) Response variables are intensities measured for H3K9me2, H3K9me3, etc.;b) The factor of variability is the cohort with 5 different levels: control, IDH1^-WT^, IDH1 mutant, IDH2^-WT^, IDH2 mutant;c) Contrasts to be tested are the ones listed under the subsection “Confirmatory analysis plan” (H3K9me2, WT vs. mutant, etc.).

Clarifications should be made for each model used in this protocol, as it is not always clear how the suggested model fits the data. The same comments hold true for the other protocols, for example the 2HG/glutamate ratios where there is only one response variable and a factor of variability with 5 levels with 2 tested contrasts.

Protocol 2:

For the 2HG/glutamate ratio, please describe the model (one factor with three levels, one contrast?).

For the qRT-PCR, is it a per-gene model? What are the response variables? Please clarify.

Protocol 3:

Again please clarify the MANOVA model (response variable, factor levels, contrasts).

---

## [Author Response]

Essential revisions: Normality and homoscedasticity will be assessed using adequate tests and methods. However what kind of transformation do the authors plan to apply in case of skewed data? Do the authors have an idea of how the data could be skewed? Similarly, which test do the authors intend to apply in case of non-normal data?

We have included the potential of a logarithmic transformation in case of skewed data in the revised manuscript. This might be the case with the 2HG/glutamate ratios that are listed in the protocols, which only show representative images, but display them on a log scale, which suggests the data might be skewed.

In the case of non-normal data, the revised manuscript includes the non-parametric tests that would be applied. When there is reason to suspect that a non-parametric test might be applied, such as a large difference in variance between the groups being tested, we have included non-parametric as well as parametric power calculations as described in response to the second question.

*Power calculations assume normally distributed data. In the opposite case non parametric tests will be used with much less power. Power calculations should be taken with caution, as they are performed according to an optimistic scenario.*

We agree and understand the limitations of using the original data we have been able to obtain, or estimate, as a basis for the sample size for the replication attempt. We also agree that some of the assumptions made are according to an optimistic scenario. However, we hope that by performing power calculations using non-parametric tests, when there is reason to believe the data would violate the assumptions of a parametric test, we will ensure the sample size used for the replication study will be suitable for a parametric test (what we aim to perform if possible) or the non-parametric test (what we quite possible will need to perform). Further, as pointed out, this still stands risk of decreased power depending on the nature of the replication data (such as if there is normal distribution) in order to perform the proposed comparisons. Additionally, we have revised the analysis plans and power calculations to reflect the proposed models and contrasts, including the potential non-parametric tests that might be performed.

Protocol 1: The authors should clarify if the MANOVA analyses are planned to analyze the results of western blots and that:

*a) Response variables are intensities measured for H3K9me2, H3K9me3, etc.;b) The factor of variability is the cohort with 5 different levels: control, IDH1^-WT^, IDH1 mutant, IDH2^-WT^, IDH2 mutant;c) Contrasts to be tested are the ones listed under the subsection “Confirmatory analysis plan” (H3K9me2, WT vs. mutant, etc.).*

*Clarifications should be made for each model used in this protocol, as it is not always clear how the suggested model fits the data. The same comments hold true for the other protocols, for example the 2HG/glutamate ratios where there is only one response variable and a factor of variability with 5 levels with 2 tested contrasts. Protocol 2: For the 2HG/glutamate ratio, please describe the model (one factor with three levels, one contrast?).*

For the qRT-PCR, is it a per-gene model? What are the response variables? Please clarify. Protocol 3: Again please clarify the MANOVA model (response variable, factor levels, contrasts).

Thank you for this suggestion. We have included the model of each proposed analysis plan, to reflect the DV, IV, and contrasts proposed. Additionally, we have revised the analysis plans and power calculations to reflect the proposed models and contrasts, included the potential non-parametric tests that might be performed.